# Energy storing bricks for stationary PEDOT supercapacitors

Hongmin Wang [1], Yifan Diao[2], Yang Lu[2], Haoru Yang[1], Qingjun Zhou[2], Kenneth Chrulski [1] &
Julio M. D'Arcy [1,2] ✉

Fired brick is a universal building material, produced by thousand-year-old technology, that throughout history has seldom served any other purpose. Here, we develop a scalable, cost-effective and versatile chemical synthesis using a fired brick to control oxidative radical polymerization and deposition of a nanofibrillar coating of the conducting polymer poly(3,4-ethylenedioxythiophene) (PEDOT). A fired brick's open microstructure, mechanical robustness and ~8 wt% $\alpha$-$Fe_2O_3$ content afford an ideal substrate for developing electrochemical PEDOT electrodes and stationary supercapacitors that readily stack into modules. Five-minute epoxy serves as a waterproof case enabling the operation of our supercapacitors while submerged underwater and a gel electrolyte extends cycling stability to 10,000 cycles with ~90% capacitance retention.

[1] Department of Chemistry, Washington University in St. Louis, St. Louis, MI 63130, USA. [2] Institute of Material Science & Engineering, Washington University in St. Louis, St. Louis, MI 63130, USA. ✉email: jdarcy@wustl.edu

Fired brick, typically used for construction and architectural esthetics, is one of the most durable materials with a 5000-year history dating back to Neolithic China[1]. This masonry building block is commonly found in various red tones and mostly comprised of fused particles of silica ($SiO_2$), alumina ($Al_2O_3$) and hematite ($\alpha\text{-}Fe_2O_3$)[2]. The red color of a brick originates from hematite, a pigment first utilized by humans 73,000 years ago[3,4] and serving today as a low-cost naturally abundant inorganic precursor for catalysts[5], magnets[6], and alloys[7]. State-of-the-art energy storage materials are also produced from hematite. For example, $FeN_x$, FeP, and $Li_5FeO_4$ are synthesized via anionic or cationic exchange for potassium-ion batteries, Zn–air batteries, pseudocapacitors, and lithium-ion batteries[8–11]; electrochemical transformation of hematite leads to FeOOH supercapacitor anodes[12].

This work is inspired by our recently published rust-assisted vapor phase polymerization[13]. Chemistries enabled by hematite provide an opportunity for developing cutting-edge functionalities on a fired brick where 8 wt% $\alpha\text{-}Fe_2O_3$ content and a 3D porous microstructure afford an ideal substrate for engineering a mechanically robust electrode. Here, we develop a supercapacitor using a brick's hematite microstructure as reactant to vapor-deposit a nanofibrillar coating of the conducting polymer poly (3,4-ethylenedioxythiophene) (PEDOT). Vapor-phase synthesis leads to PEDOT coatings exhibiting a high electronic conductivity[14] and facile charge transfer, making it an ideal route for producing electrodes[15]. This synthesis utilizes a brick's open microstructure and thermal stability to permeate acid and monomer vapor through its pores at 160 °C to control $\alpha\text{-}Fe_2O_3$ dissolution and $Fe^{3+}$ hydrolysis with concomitant oxidative radical polymerization.

A symmetric brick-based supercapacitor shows an areal capacitance of 1.60 F cm$^{-2}$ and energy density of 222 $\mu$Wh cm$^{-2}$ at a current density of 0.5 mA cm$^{-2}$. This two-electrode-based measurement is collected using 1 M $H_2SO_4$ aqueous electrolyte under 1 V operating voltage window. To mimic a "brick-mortar-brick" structure, a supercapacitor is modified using a quasi-solid-state electrolyte (poly(vinyl alcohol)/1 M $H_2SO_4$) that also plays the role of binder and separator. Our devices are water-resistant because they are coated with an epoxy encapsulating layer that protects them enabling charge storage at temperatures between −20 and 60 °C. A supercapacitor is stable in ambient conditions undergoing 10,000 charge–discharge cycles with ~100% coulombic efficiency and ~90% capacitance retention. Moreover, a supercapacitor brick module is produced reaching a 3.6 V voltage window by connecting three devices in series.

## Results

### Conversion of a fired brick's $\alpha\text{-}Fe_2O_3$ to a PEDOT coating.

Deposition of PEDOT nanofibers is initiated by dissolving $\alpha$-$Fe_2O_3$ at 160 °C with HCl vapor; this process liberates $Fe^{3+}$ ions, promotes hydrolysis and initiates precipitation of colloidal 1D FeOOH nuclei (Fig. 1a). As previously reported, partially dissolved FeOOH nuclei serving as templates oxidize 3,4-ethylenedioxythiophene (EDOT) monomer vapor and control oxidative radical polymerization[13]. Control of both stoichiometry and reaction time leads to PEDOT coatings with different thicknesses (Fig. 1b). We advance previous findings by demonstrating here that the synthesis utilizes two potential polymerization initiators, i.e., oxidant ($Fe^{3+}$) and acid (HCl) where the former leads to oxidative radical polymerization and the latter, to acid-catalyzed polymerization (Supplementary Fig. 1). We readily control the polymerization mechanism as an acid-catalyzed polymerization typically produces nonconductive oligomers stemming from active chain termination[16]. Only PEDOT synthesized via

oxidative radical polymerization exhibits long conjugation length, ordered chain packing, low electrical resistance, as well as high chemical and physical stability[13,14,17,18].

In our polymerization mechanism, the acid concentration determines both dissolution rate and synthetic pathways. A high $H^+$ concentration facilitates the liberation of $Fe^{3+}$ as well as oxidative radical polymerization while promoting acid-catalyzed polymerization that consumes EDOT and results in PEDOT of low conjugation length (Fig. 1c). To investigate the effect of $H^+$ concentration, different volumes of concentrated HCl are added to the reactor and the HCl vapor concentration is calculated by dividing moles of HCl with reactor volume (assuming total evaporation). Using a HCl vapor concentration less than 4.8 mM leads to an incomplete reaction because both oxidative radical polymerization and acid-catalyzed polymerization are impeded (Supplementary Fig. 2a). Increasing concentration to 14 mM liberates $Fe^{3+}$ and promotes oxidative radical polymerization resulting in PEDOT of low electrical resistance, whereas concentrations above 14 mM activate the acid-catalyzed polymerization pathway resulting in uncontrolled reactions (Supplementary Fig. 2b). When oxidative radical polymerization dominates, the EDOT vapor concentration determines the thickness of a polymer coating and its electrical resistance (Supplementary Fig. 2a, c).

Synthesis starts when a brick and chemical reactants are heated together in a sealed vessel; chemical and physical changes on a brick are monitored by collecting sample aliquots at different time intervals. Notably, there is no color change on a brick's red surface during the first 4 h of reaction because brick dissolution is the rate-limiting step in our mixed synthetic mechanism that consists of (1) evaporation, (2) dissolution, (3) hydrolysis, and (4) polymerization. A blue PEDOT coating is visible on a brick 4 h after initiating a reaction and its thickness increases inversely proportional with electrical resistance until the end of reaction at 14 h. An extended polymerization time increases the polymer coating's two-point probe electrical resistance (Supplementary Fig. 2d) because PEDOT loses dopant during heating[18]; fortunately, post-synthetic doping lowers the electrical resistance. Our synthesis produces a 400 $\mu$m thick nanofibrillar PEDOT coating (2.8 wt%) exhibiting 2$\Omega$ two-point probe electrical resistance and nanofibers characterized by a ~30 $\mu$m length and ~190 nm diameter (Supplementary Fig. 3a–d). Polymer, purified by repeated rinses in methanol, is comprised of S, C, O, and doped by Cl$^-$ in situ during polymerization as shown in energy-dispersive X-ray spectra (Supplementary Fig. 3e).

Our synthesis is general and applicable to different types of bricks. Here, three types of bricks (type 1–3) with different gravel ($SiO_2$) sizes and porosities are investigated (Fig. 2). Type 1 brick shows the most open microstructure (Fig. 2a) that facilitates reagent vapor diffusion, therefore utilized for the synthesis study in Fig. 1 and Supplementary Figs. 1–3. Powder X-ray diffraction of pulverized type 1 brick shows that $SiO_2$ is the major phase while $\alpha\text{-}Fe_2O_3$ and $Al_2O_3$ are minor phases (Fig. 2c). These three types of bricks possess similar inorganic composition and concentration as shown by powder X-ray diffraction patterns (Fig. 2d). Porosity differences between brick types stem from various gravel sizes and manufacturing variables such as water content before sintering, sintering temperature and duration. Vapor-phase synthesis produces a contiguous polymer coating over the entire brick surface because dissolution generates an aqueous $Fe^{3+}$ layer that coats inert gravel sites. Under identical reaction stoichiometries and time, PEDOT nanofibers of high aspect-ratio coat all types of bricks homogeneously (Fig. 2b). Our polymer coating technology is scalable (Fig. 2e) and patternable (Supplementary Fig. 4a) as demonstrated using type 1 brick. Nanofibrillar PEDOT coatings on all types of bricks show linear

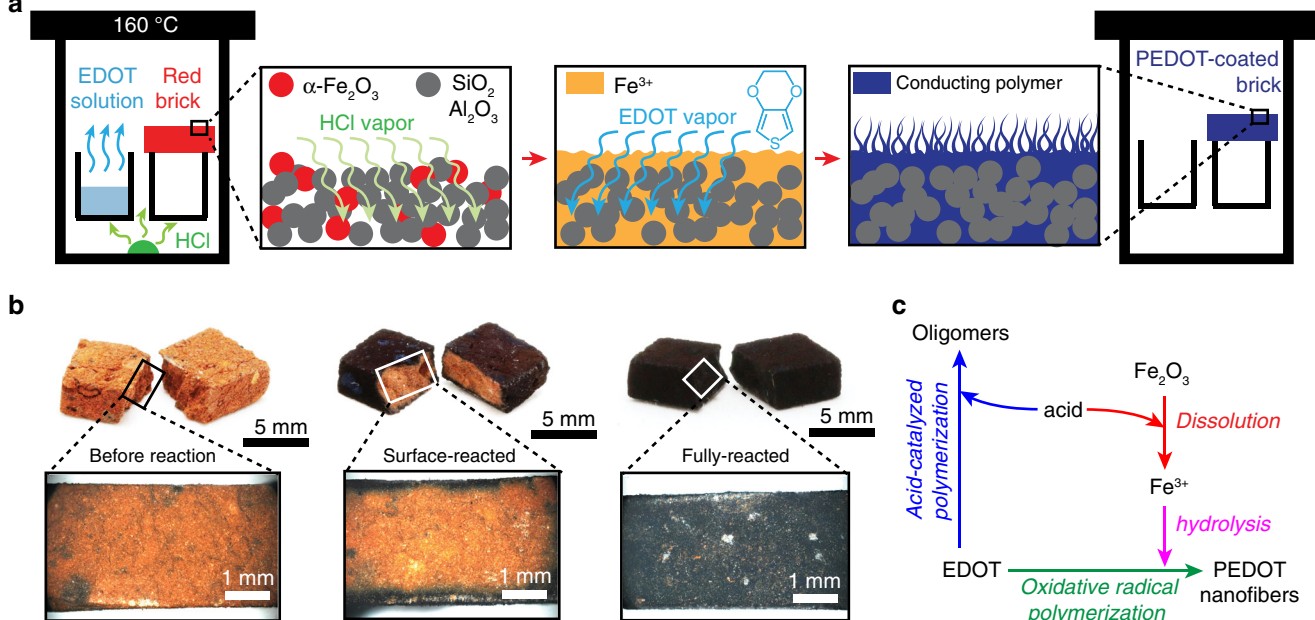

**Fig. 1 Deposition of a nanofibrillar PEDOT coating on brick. a** In a one-step reaction, a brick's α-Fe$_2$O$_3$ microstructure is partially dissolved by acid vapor to liberate Fe$^{3+}$, promote hydrolysis and precipitation of FeOOH spindles that control oxidative radical polymerization. As previously reported, monomer vapor reacts with partially dissolved FeOOH nuclei resulting in preferential directional growth of high aspect ratio PEDOT nanofibers[13]. **b** The thickness of a PEDOT coating is controlled by reaction time and stoichiometry enabling a reaction to generate surface-polymerized PEDOT-coated bricks (core/shell architecture) or fully polymerized bricks (monolithic PEDOT architecture). **c** A reaction diagram shows the competition between acid-catalyzed polymerization and oxidative radical polymerization mechanisms present in our reactions.

current–voltage curves with slopes that indicate ohmic behaviors and similar electrical resistances (~7Ω) (Fig. 3a). Note that a two-point probe multimeter measurement leads to a lower value (2Ω) because of a wide tip diameter that lowers contact resistance.

In order to study the effect of polymer coatings on the porosity of bricks, water absorption experiments are carried out before and after synthesis. A PEDOT-coated type 1 brick absorbs the most weight of water among the three types of polymer-coated bricks due to its open pore structure and large pore size and volume (Fig. 2a). A nanofibrillar PEDOT coating on type 1 brick exhibits minimal delamination via Scotch tape tests whereas a coating of the commercial product PEDOT:poly(styrenesulfonate) peels off readily and completely. Note that both type 2 and type 3 bricks show partial delamination (Fig. 3c) of our polymer coating because of a semi-closed brick microstructure (Fig. 2a) that impedes vapor reactant diffusion (Fig. 3e, f). Cross-sectional scanning electron micrographs reveal poor PEDOT interpenetration into the brick's pores, this leads to a surface localized coating with minimal anchoring that is prone to delamination. Fortunately, type 1 brick's open microstructure enables reagent vapor diffusion producing in situ PEDOT grafting; a nanofibrillar polymer coating is embedded as a network throughout pores resulting in strong adhesion (Fig. 3d). The addition of α-Fe$_2$O$_3$ particles enables deposition of PEDOT coatings on customized substrates such as concrete road pavers (Supplementary Fig. 4b) and Portland-based white concrete (Supplementary Fig. 4c).

**Nanofibrillar PEDOT-coated brick electrochemical electrode.** We present a summary of geometries and mass loadings for electrodes and devices in Supplementary Table 1 while areal, gravimetric, and volumetric capacitances, energy and power densities are shown in Supplementary Table 2 (calculations in detail are included in "Supplementary Methods" section). We choose areal metrics for evaluating our electrodes and devices because area is a practical parameter for evaluating our wall-like

brick-based supercapacitor and it is directly related to the thin polymer coating (Supplementary Fig. 5) present on a wall. Areal metrics facilitate the estimation of capacitance and energy density delivered by a brick wall (Supplementary Discussion).

A 1 cm × 0.5 cm × 0.28 cm nanofibrillar PEDOT-coated type 1 brick (0.14 cm$^3$) weighing 249 mg and carrying 6.97 mg PEDOT is fabricated into an electrode with one 1 cm × 0.5 cm face exposed (Fig. 4a). When calculating areal metrics, we use the area of the face exposed to electrolyte (0.5 cm$^2$). The electrode exhibits a quasi-rectangular shaped three-electrode cyclic voltammogram and a capacitance of 2.61 F cm$^{-2}$ (187 F g$^{-1}$ based on PEDOT's mass) at 2 mV s$^{-1}$ in 1 M H$_2$SO$_4$ (Fig. 4b). The low scan rate of 2 mV s$^{-1}$ enables enough time for charge transfer in the thick electrode to calculate maximum capacitance and energy density. We summarize capacitances obtained at scan rates from 2 to 100 mV s$^{-1}$ in Supplementary Fig. 6a. The Fe$^{3+}$/Fe$^{2+}$ redox pair at 0.37 V and 0.49 V (vs. Ag/AgCl) arise due to iron species in brick and disappears as scan rate increases to 25 mV s$^{-1}$ because Faradaic processes in PEDOT are faster than those occurring from solid Fe$_2$O$_3$ (Fig. 4c inset)[15,19]. Rate performance tests demonstrate capacitive behavior as scan rate increases to 100 mV s$^{-1}$; however, due to limited charge transport the curve changes to fusiform shape[20] (Fig. 4c). Capacitance is also dependent on the aqueous electrolyte and sulfuric acid leads to greater capacitance (1.64 F cm$^{-2}$) than sodium sulfate (0.878 F cm$^{-2}$) at 25 mV s$^{-1}$ (Fig. 4d). This drastic difference is plausibly due to higher ionic mobility in H$^+$ (36.23 × 10$^{-8}$ m$^2$ V$^{-1}$ s$^{-1}$) vs. Na$^+$ (5.19 × 10$^{-8}$ m$^2$ V$^{-1}$ s$^{-1}$) or a lower electrical resistance caused by doping at low pH[14]. We probed this behavior further using electrochemical impedance spectroscopy. Nyquist plots and an equivalent circuit diagram demonstrate a significantly lower ion diffusion resistance for H$_2$SO$_4$ (1.7Ω) vs. Na$_2$SO$_4$ (4.6Ω) and minimal change in electrode material electrical resistance at low pH (Fig. 4e). To test the effect of an acidic electrolyte on a brick's inorganic species, we pulverize and homogenize the entire

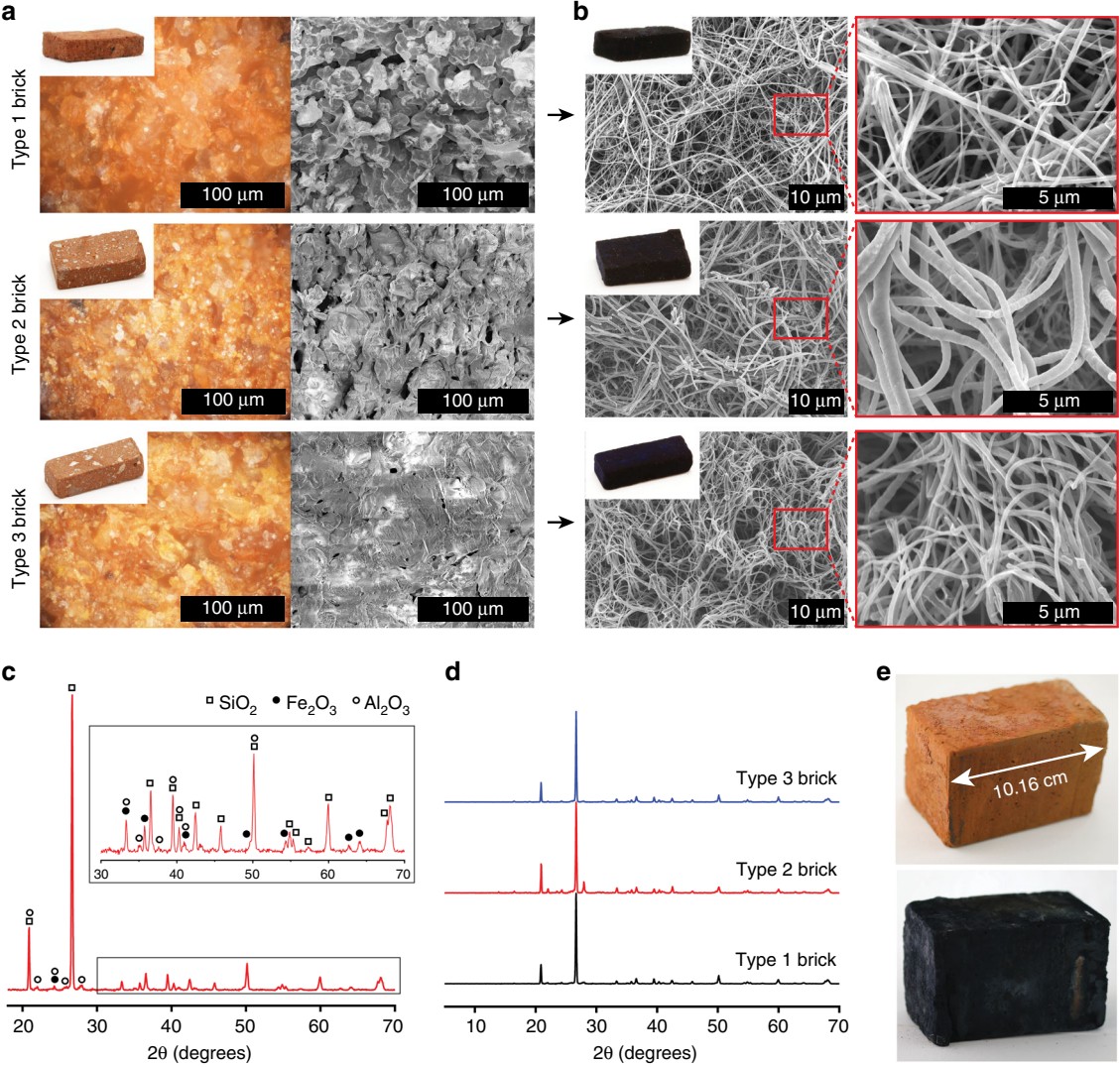

**Fig. 2 Nanofibrillar PEDOT-coatings on different types of bricks. a** Three types of red fired bricks are utilized for synthesis. All types show homogeneous red-orange color under the optical microscope, indicating uniform α-Fe$_2$O$_3$ distribution. White spots on bricks (left top inset) show the gravel (SiO$_2$) sizes increasing from type 1 to type 3 brick. Scanning electron micrographs exhibit different porosity for these three types of bricks with pore sizes decreasing from type 1 to type 3. **b** After synthesis, all types of bricks are homogeneously coated by PEDOT nanofibers of similar aspect ratio. **c** Powder X-ray diffraction pattern for type 1 brick shows a composition comprised of SiO$_2$, α-Fe$_2$O$_3$, and Al$_2$O$_3$. **d** Type 2 and type 3 bricks show similar patterns and crystalline components. The sharper and stronger peak at $2\theta = 28°$ from type 2 brick is due to its higher Al$_2$O$_3$ content or crystallinity. **e** Synthesis is scalable to decimeter-sized bricks.

polymer-coated brick electrode using mortar and pestle. Powder X-ray diffraction is carried out before and after electrodes are cycled in 1 M H$_2$SO$_4$ using sequential scan rates of 2, 5, 10, 25, 50, and 100 mV s$^{-1}$ (10 cycles each). Diffraction patterns remain unchanged demonstrating that most inorganic species in a brick remain unaffected by synthesis and electrochemical cycling (Fig. 4f).

To quantify dissolution of a brick's α-Fe$_2$O$_3$ and Al$_2$O$_3$ by 1 M H$_2$SO$_4$, we perform inductive coupled plasma mass spectrometry on the electrolyte after cycling experiments (Supplementary Fig. 6b, c). For a precise calculation, we control electrolyte volume (5 mL) and electrode mass (249 mg). Analysis shows negligible concentrations of Fe (4.44 µg mL$^{-1}$) and Al (1.97 µg mL$^{-1}$) in the electrolyte after cycling, these concentrations are equivalent to mass losses (based on the entire brick) of 0.0127 and 0.0075 wt% for α-Fe$_2$O$_3$ and Al$_2$O$_3$, respectively. These results confirm that inorganic species in a brick are preserved after electrochemical cycling.

**Symmetric nanofibrillar PEDOT-coated brick supercapacitor.** Two nanofibrillar PEDOT-coated bricks (1 cm × 0.5 cm × 0.28 cm) serve as electrodes in a symmetric supercapacitor (1 cm × 0.5 cm × 0.5625 cm) using 1 M H$_2$SO$_4$ aqueous electrolyte (Fig. 5a upper row and Supplementary Fig. 7a). A device consists of a volume (0.28 cm$^3$) encompassing two PEDOT-coated bricks and a separator and its total mass (499 mg) includes 13.94 mg of PEDOT. Nyquist plot shows an aggregated internal resistance of 3Ω and a line with a ~45° slope between semicircle and low-frequency domain (Warburg region) (Supplementary Fig. 7b). This line is characteristic of thick electrodes where a tortuous path stifles ion diffusion. At the low-frequency region, the curve tends to form an arc because of Nernst diffusion impedance of H$^+$ and SO$_4^{2-}$ in the electrolyte[21]; this plausibly stems from an increased ion diffusion distance caused by a separation gap between anode and cathode. Cyclic voltammogram shows a quasi-rectangular shape between 0 and 1 V (collected at 2 mV s$^{-1}$) leading to a device areal capacitance of 1.59 F cm$^{-2}$

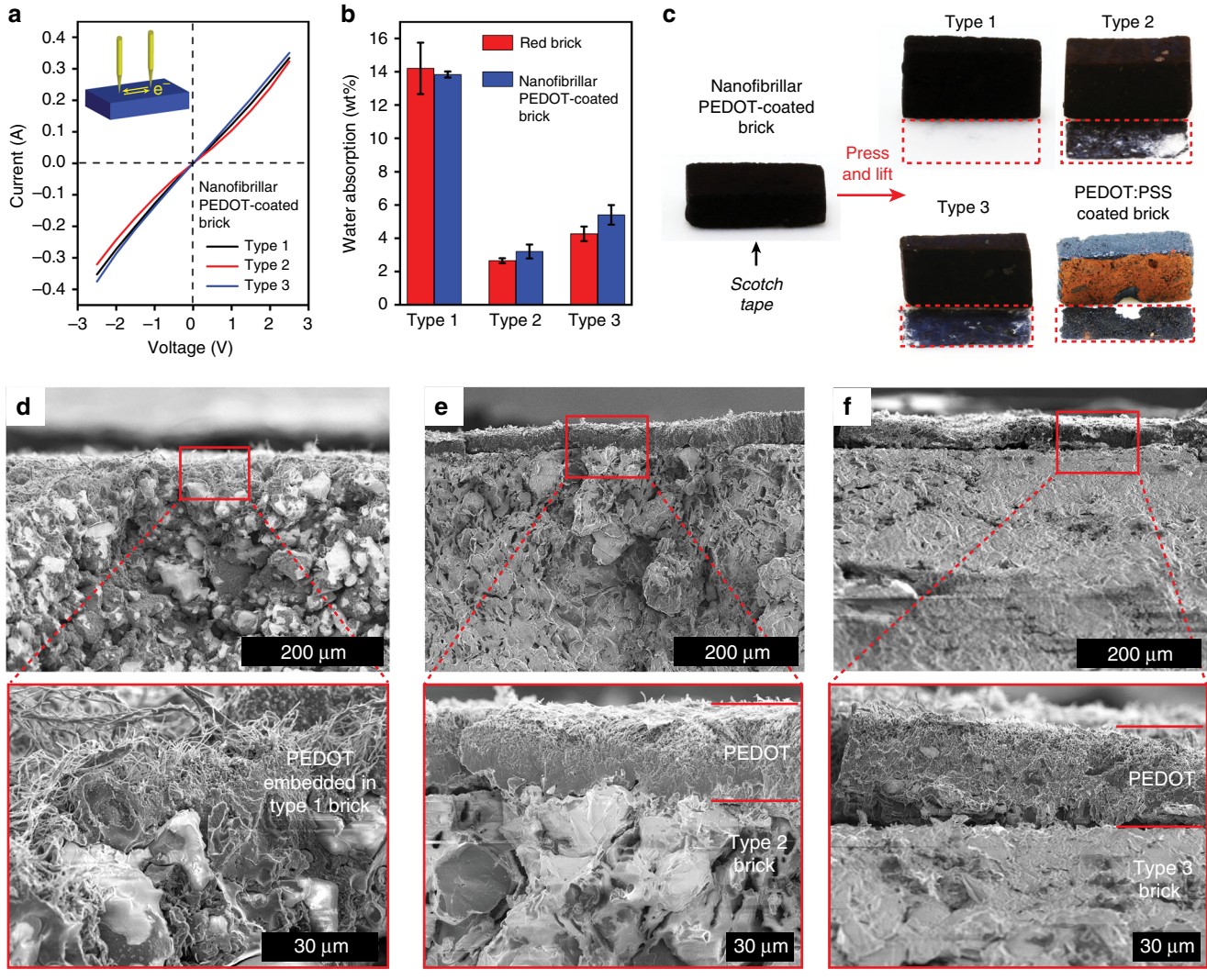

**Fig. 3 Analysis of PEDOT coatings on various bricks. a** Linear current–voltage curves show ohmic behaviors of nanofibrillar PEDOT coatings possessing comparable resistances. **b** Water absorption experiments on bricks enables studying their open pore structure and all bricks show insignificant changes in water absorption after PEDOT coatings with type 1 brick absorbing the most water due to a higher porosity. **c** A nanofibrillar PEDOT coating exhibits superior adhesion on type 1 brick versus other types during Scotch tape tests. Unfortunately, the commercial product PEDOT:poly(styrene sulfonate) delaminates completely after coated on type 1 brick. **d** Cross-sectional scanning electron micrographs show the embedded PEDOT network in type 1 brick leading to robust PEDOT adhesion. Type 2 (**e**) and type 3 (**f**) bricks show distinct boundaries between PEDOT and brick that are prone to delamination.

calculated using the electrode area directly in contact with separator (0.5 cm²) (Fig. 5b, black curve and Supplementary Fig. 7c). In our device, a nanofibrillar PEDOT coating covers all six faces of a brick, two of the larger faces (1 cm × 0.5 cm) are directed towards each other and during cycling, ions travel through the inner brick pores resulting in an electrochemical contribution by the other faces. The areal metrics of this aqueous electrolyte supercapacitor are thus qualitative.

Our supercapacitor possesses low internal resistance resulting in a low IR drop (0.01 V) during galvanostatic charge-discharge experiments at 0.5 mA cm⁻² current density in a 1 V window (Supplementary Fig. 7d). These curves demonstrate a device areal capacitance of 1.60 F cm⁻² (2.84 F cm⁻³ for volumetric) as well as areal energy and power densities of 222 and 0.25 mW cm⁻², respectively (394 µWh cm⁻³ and 0.44 mW cm⁻³ for volumetric). High power density (12.5 mW cm⁻²) is obtained at a current density of 25 mA cm⁻² albeit with lowered capacitance (0.706 F cm⁻²) and energy density (98 µWh cm⁻²) because ion transport in our thick electrode is limited (shown by high IR drop of 0.4 V). Our device works in an extended voltage window

(1.2 V) resulting in cyclic voltammograms and galvanostatic charge–discharge curves that retain shape (Supplementary Fig. 7e, f). However, the increased IR drop (0.07 V) at 1.2 V decreases coulombic efficiency as well as long term cycling stability leading to lower capacitance versus 1 V (Supplementary Fig. 7g, h). The energy densities at 1.2 V are higher than at 1 V as shown in a Ragone plot (Supplementary Fig. 7i) because a wider voltage window allows a device to store more charge (Supplementary Eq. 7 from "Supplementary Methods" section). Connecting three devices in series increases the voltage window to 3.6 V, this also triples the internal resistance and reduces output current to one-third (Supplementary Fig. 8a). A tandem device reaches an output voltage of 2.685 V (charged at 4.5 V for 15 s) and lights a white light-emitting diode for 11 min. The output voltage decreases to the same level as the light-emitting diode's turn-on voltage (2.546 V) after discharging for 214 s (Supplementary Fig. 8b–d and Supplementary Movie 1).

**Quasi-solid-state PEDOT-coated brick supercapacitor**. To minimize electrolyte leakage, we develop a symmetric supercapacitor

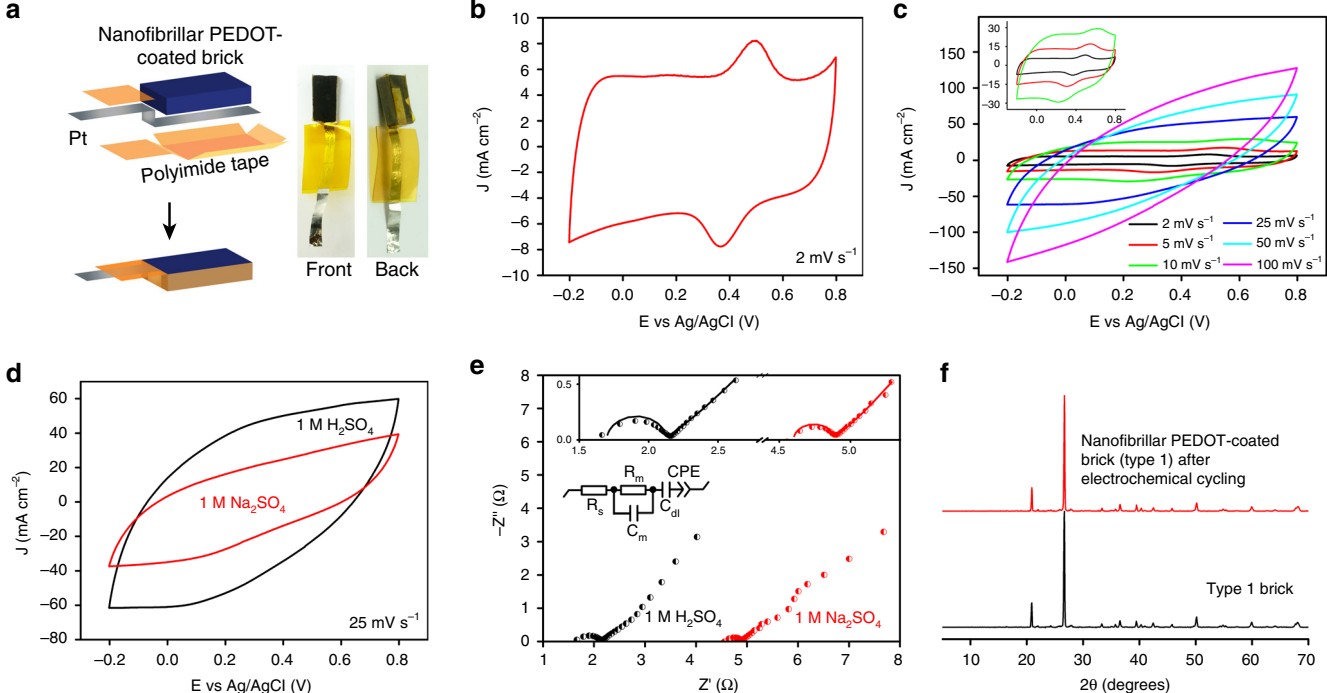

**Fig. 4 Three-electrode characterization of nanofibrillar PEDOT-coated bricks. a** Nanofibrillar PEDOT-coated type 1 brick (1 cm × 0.5 cm × 0.28 cm) is connected to a Pt current lead using polyimide tape that exposes a 1 cm × 0.5 cm face to the electrolyte. **b** Cyclic voltammogram at 2 mV s$^{-1}$ shows a quasi-rectangular shape stemming from PEDOT's capacitive behavior with $Fe^{3+}/Fe^{2+}$ redox pair peaks at 0.37 and 0.49 V (vs. Ag/AgCl). **c** The $Fe^{3+}/Fe^{2+}$ redox peaks disappear when scan rate increases to 25 mV s$^{-1}$. **d** The electrode shows a smaller curve area for 1 M $Na_2SO_4$ electrolyte compared to 1 M $H_2SO_4$, indicating lower capacitance. **e** Nyquist plots collected using different electrolytes (inset shows equivalent circuit diagram) show fitted (solid lines) versus experimental data (segregated points). **f** Powder X-ray diffraction shows identical patterns for pristine type 1 brick vs. PEDOT-coated type 1 brick after cycling in 1 M $H_2SO_4$ between −0.2 and 0.8 V (vs. Ag/AgCl) at scan rates of 2, 5, 10, 25, 50, and 100 mV s$^{-1}$ (10 cycles each).

(1 cm × 0.5 cm × 0.63 cm) using a poly(vinyl alcohol)/1 M $H_2SO_4$ gel that binds PEDOT-coated bricks and serves as electrolyte and separator (Fig. 5a lower row and Supplementary Fig. 9a). The gel electrolyte layer (0.7 mm thick) prevents bricks (1 cm × 0.5 cm × 0.28 cm) from short-circuiting and leads to enhanced adhesion between electrodes. This sandwich-type supercapacitor is 0.32 cm$^3$ in volume, 518 mg in weight (includes mass of electrodes and electrolyte) containing 13.94 mg of PEDOT. In a tensile test, our electrode–gel–electrode structure withstands a shearing force equal to 1000 times the device's weight[20] (Supplementary Fig. 9b). Intimate contact between gel and PEDOT nanofibers enhances charge transfer resulting in low internal resistance (2.5 Ω) and a linear Nyquist plot (Supplementary Fig. 9c). Device areal capacitance (0.868 F cm$^{-2}$) and areal energy density (121 µWh cm$^{-2}$) originating from PEDOT-coated brick electrodes are calculated from galvanostatic charge–discharge curves (1.38 F cm$^{-3}$ and 192 µWh cm$^{-3}$ for volumetric, collected at 0.5 mA cm$^{-2}$); cyclic voltammograms also show capacitive behavior (Fig. 5b, c, Supplementary Fig. 9d–g). The absence of redox peak from cyclic voltammogram indicates a minimal contribution from the α-$Fe_2O_3$ present in a brick to energy storage. A gel electrolyte leads to 50% lower areal capacitance and energy density than an aqueous electrolyte because of stifled gel permeation kinetics throughout the electrode preventing access of ions to all non-sandwiched PEDOT-coated faces (Fig. 5a, Supplementary Fig. 9h).

Outdoor exposure is inevitable for a stationary supercapacitor and epoxy encapsulation affords a cost-effective, mechanically robust and waterproof housing. An epoxy-coated supercapacitor retains ~90% of original capacitance and exhibits ~100% coulombic efficiency after 10,000 charge–discharge cycles (collected at 25 mA cm$^{-2}$) (Fig. 5d, red curves). This 5-min epoxy

coating prevents water evaporation from the gel's hydrated ionic percolation network[22] (Supplementary Fig. 10a) enabling 10,000 charge–discharge cycles at 5 mA cm$^{-2}$ (640 h of continuous operation) with ~87% capacitance retention (Fig. 5d, black curves). Gel electrolyte and encapsulation enables operation at temperatures between −20 and 60 °C (this range covers most outdoor temperatures) as shown by cyclic voltammograms and Nyquist plots (Supplementary Fig. 10b, c). The capacitance increases proportionally with temperature due to enhanced ionic transport[22]; PEDOT remains capacitive after repeated reversible heating-cooling cycles (Supplementary Fig. 10d). Temperatures below −20 °C or above 60 °C cause significant freezing or evaporation of water from the gel electrolyte leading to unstable electrical performance and breakage of epoxy seal.

Epoxy renders a stationary supercapacitor module waterproof. Three quasi-solid-state supercapacitors are sealed in epoxy pucks (Supplementary Fig. 9a) and connected in series by electrical wires. This entire module (including connection points) is coated by another layer of epoxy that leaves only cathode and anode exposed. After epoxy curing, the entire module is immersed in water except for two electrode-leads and is electrochemically cycled (electrical circuit shown in Fig. 5e inset and Supplementary Fig. 10e). The module's cyclic voltammograms (Fig. 5e) collected while immersed demonstrate stable behavior identical to pre-immersion tests. A device charges to 3 V in 10 s while immersed in water and lights up a green light-emitting diode (2.155 V forward voltage) for ~10 min (Supplementary Fig. 10e, f). A gel electrolyte and our deposition technology enable scale up as demonstrated by connecting six large nanofibrillar PEDOT-coated brick electrodes (2 cm × 1 cm × 1 cm) in series resulting in a supercapacitor module that charges to 3 V in 5 s readily lighting up a green light-emitting diode (Fig. 5f, Supplementary Fig. 11).

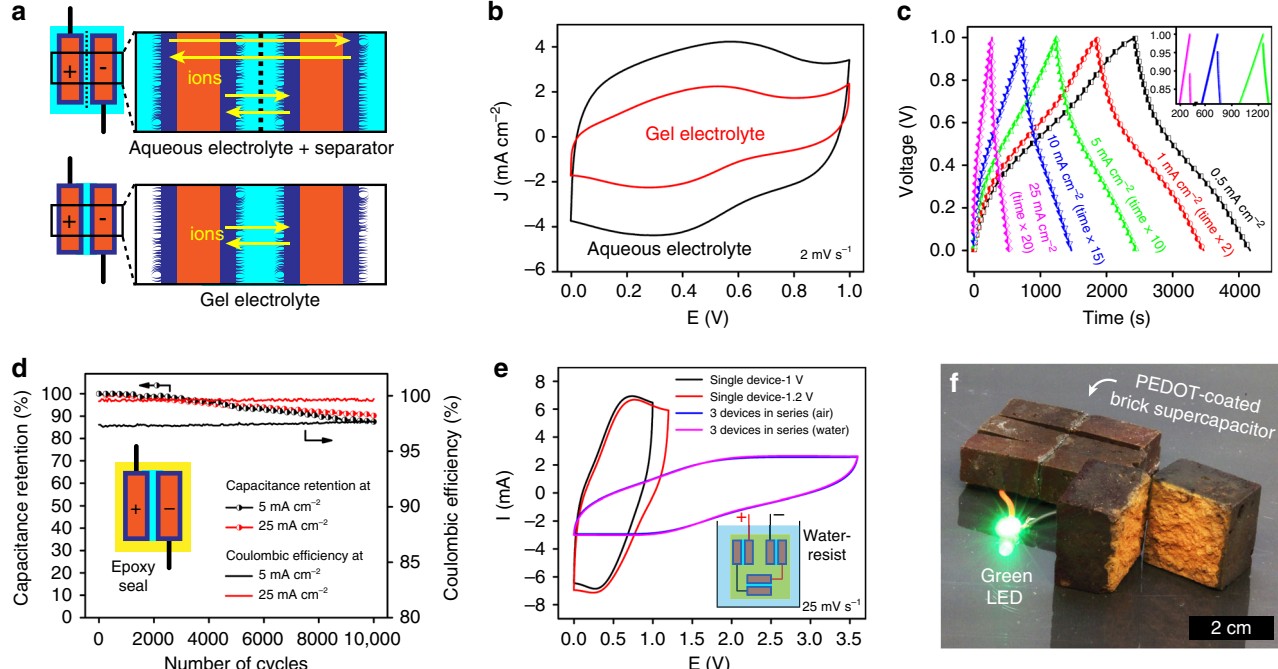

**Fig. 5 Nanofibrillar PEDOT-coated bricks for supercapacitors. a** Schematic illustration of aqueous electrolyte supercapacitor and quasi-solid-state supercapacitor shows different charge storage sites. The black dash line in the aqueous electrolyte device represents a separator. **b** Cyclic voltammograms for symmetric supercapacitors using 1 M $H_2SO_4$ aqueous electrolyte and poly(vinyl alcohol)/$H_2SO_4$ gel electrolyte. **c** Galvanostatic charge–discharge profiles for quasi-solid-state device at current densities ranging between 0.5 and 25 mA cm$^{-2}$; curves at 1, 5, 10, and 25 mA cm$^{-2}$ are horizontally expanded 2×, 10×, 15×, and 20×, respectively. Inset shows IR drop at current densities of 5, 10, and 25 mA cm$^{-2}$. **d** Quasi-solid-state supercapacitor charge-discharge curves after 10,000 cycles at 5 and 25 mA cm$^{-2}$ exhibit 87% and 90% capacitance retention, respectively (coulombic efficiency is ~100%). **e** Cyclic voltammograms for a single quasi-solid-state supercapacitor are collected at voltage windows of 1 and 1.2 V; a tandem device (comprised of three supercapacitors connected in series) withstands a 3.6 V window. The tandem device is waterproof after coated by epoxy and exhibits a stable cyclic voltammogram. **f** Photograph shows a supercapacitor module lighting up a green light-emitting diode. This tandem device (4 cm × 3 cm × 1 cm) contains three supercapacitors (4 cm × 1 cm × 1 cm) connected in series; the core–shell structure of an electrode is also shown.

Notably, a brick wall constructed using our nanofibrillar PEDOT-coated bricks holds the potential to deliver a maximum device capacitance of 11.5 kF m$^{-2}$ and an energy density of 1.61 Wh m$^{-2}$ (Supplementary Fig. 12; Supplementary Discussion).

## Discussion

This proof-of-concept work demonstrates how to store energy on the surface of a common brick using α-$Fe_2O_3$ as an oxidant precursor to control oxidative radical polymerization and conformally deposit a capacitive nanofibrillar PEDOT coating from the vapor phase. A brick's structural stability and open microstructure result in mechanically robust PEDOT-coated brick electrodes, that when connected in series and coated with epoxy, produce a stable stationary waterproof supercapacitor module. Our supercapacitor technology adds value to a "dirt-cheap" construction material and demonstrates a scalable process affording energy storage for powering embedded microdevices in architectural applications that utilize fired brick.

## Methods

**Materials**. Chlorobenzene (99%), 3,4-ethylenedioxythiophene (97%), poly(vinyl alcohol) ($M_w$ 89,000–98,000, 99+% hydrolyzed), methanol (≥99.8%), and hydrochloric acid (37%) are purchased from Sigma-Aldrich; sulfuric acid (AR) is purchased from Macron. The PEDOT:PSS solution (Clevios PH 1000) is purchased from Heraeus company. All chemicals are used without further purification. Platinum foil (0.025 mm thick, 99.9%) is purchased from Alfa Aesar and utilized for engineering electrode leads and Celgard 3501 membrane is used as a separator. Fired bricks are purchased from local hardware stores: The Home Depot Inc. (type 1 brick, https://www.homedepot.com, Internet #100323015, Model #RED0126MCO, Store SKU #393134), Lowe's Inc. (type 2 brick, https://www.lowes.com, Item # 10298, Model # 600370) and Menards Inc. (type 3 brick, https://www.menards.com, Model

Number: 1441901, Menards® SKU: 1441901). Road pavers are purchased from Menards Inc. (https://www.menards.com, Model Number: 1793028, Menards® SKU: 1793028) and all construction materials used for developing electrodes are cut using a diamond saw. Red cement color (α-$Fe_2O_3$ particles) produced by NewLook Inc. is purchased from The Home Depot Inc. (https://www.homedepot.com, Internet #203858654, Model #CC1LB105) and serves as an oxidant source for developing chemical syntheses. Materials for making concrete (purchased from The Home Depot Inc., https://www.homedepot.com) include commercial-grade Quikrete Portland cement (Type I/II) (Internet #100318486, Model #100700, Store SKU #616788), Quikrete all-purpose sand (Internet #100318450Model # 115251Store SKU #137263) and Pavestone multi-purpose patio/paver base (Internet #100580973, Model #98001, Store SKU #208618).

**Characterization**. Scanning electron micrographs and energy-dispersive X-ray spectra are collected with a JEOL 7001LVF FE-SEM. Two-point probe resistance measurements are carried out using a Fluke 177 True RMS digital multimeter with 3 mm distance between two probes. Thermogravimetric analysis is conducted on a Discovery TGA (TA Instruments). Cyclic voltammetry, galvanostatic charge–discharge and electrochemical impedance spectroscopy are performed in a BioLogic VMP3 multipotentiostat. For electrochemical impedance spectroscopy, the sinusoidal disturbance is 10 mV with frequencies scanned between 100 kHz and 0.1 Hz. A Nyquist plot shows real impedance $Z'$ vs. imaginary impedance $-Z''$ under a sinusoidal disturbance at the open circuit potential. Fitting of Nyquist plot using an equivalent circuit diagram contains solution resistance ($R_s$), electrode material resistance ($R_m$), material capacitance ($C_m$), double-layer capacitance ($C_{dl}$), and constant phase element (CPE). Here, $R_s$ reflects the electrolyte ionic mobility and $R_m$ represents the electrical resistance of the electrode. Powder X-ray diffraction spectra of brick powders pulverized by mortar and pestle are obtained in a Bruker d8 Advance X-ray diffractometer at room temperature, with a Cu Kα radiation source ($\lambda = 1.5406$ Å) and LynxEyeXe detector. The sample holder is a background-free silicon disk rotating at 30 rpm when collecting data with a 0.02° scan step at 40 kV and 40 mA. Current–voltage tests are performed on a 3D printed two-point probe station with two gold probes separated by 2 mm[23]. Water absorption experiments are performed as described in ASTM C67/C67M-18 except brick samples are 1 cm × 0.5 cm × 0.28 cm in size. Inductively coupled plasma mass spectrometry is performed on a Perkin Elmer

ELAN DRC II ICP–MS. Samples for testing are obtained from electrolytes (5 mL 1 M $H_2SO_4$) after three-electrode cyclic voltammetry experiments and are diluted to 1/100 with Mili-Q water before analyses. External calibration curves are obtained with IV-ICPMS-71A standard solution purchased from Inorganic Ventures, Inc.

**Preparation of poly(vinyl alcohol)/$H_2SO_4$ gel electrolyte**. The gel electrolyte is formulated using 1 g of poly(vinyl alcohol) powder dissolved in 10 mL deionized water under vigorous stirring at 90 °C and cooled to around 50 °C. Dropwise addition of 1 g of concentrated $H_2SO_4$ (1 M) is then carried out by pipetting acid on the inner wall under vigorous stirring to prevent carbonization of poly(vinyl alcohol). Stirring minimizes localized heating and is carried out for 1 h resulting in a homogeneous, translucent, and colorless solution.

**Synthesis of nanofibrillar PEDOT coating on a brick**. Brick is cut using a diamond saw (±0.03 cm error) into the following 4 different sizes: 1.00 cm × 0.50 cm × 0.28 cm (for studying synthesis and electrochemistry), 1.27 cm × 1.27 cm × 0.20 cm (for patterning), 2.00 cm × 1.00 cm × 1.00 cm (for scaled-up supercapacitor) and 10.16 cm × 6.77 cm × 5.72 cm (for scaled-up synthesis). A brick is thrice washed with deionized water to remove surface dust then dried at 160 °C for 1 h and cooled to room temperature.

The syntheses of all types of 1 cm × 0.5 cm × 0.28 cm brick are performed in a 25 mL Teflon-lined stainless-steel autoclave as shown in Fig. 1a. A brick is placed on a glass reservoir then 200 μL of a 0.85 M EDOT/chlorobenzene solution is loaded in a separate glass reservoir and 30 μL of 12 M HCl is directly injected in the Teflon liner. The reactor is closed and introduced into an oven at 160 °C for 14 h. The product is washed thrice with excess methanol and dried at room temperature before carrying out tests. Similarly, the synthesis of a 2 cm × 1 cm × 1 cm brick is performed in a 125 mL Teflon-lined stainless-steel autoclave using 600 μL of a 0.70 M EDOT in chlorobenzene solution and 75 μL of 12 M HCl.

A brick (1.27 cm × 1.27 cm × 0.20 cm) is patterned using a polyimide tape mask. Synthesis is carried out at 150 °C for 14 h in a 125 mL Teflon-lined stainless-steel autoclave containing 1 mL of a 0.85 M EDOT solution in chlorobenzene and 0.6 mL of 12 M HCl (Supplementary Fig. 4a).

Scaling is carried in a glass reactor (12.30 cm × 8.55 cm × 11.30 cm) with a large brick (10.16 cm × 6.77 cm × 5.72 cm). The reaction is carried out using 15 mL of 12 M HCl and 15 mL of a 0.85 M EDOT in chlorobenzene solution at 150 °C for 6 h (Fig. 2e).

We produce a PEDOT coating on concrete by applying α-$Fe_2O_3$ particles to a concrete surface (Supplementary Fig. 4c). This composite is produced by mixing sand, stone, Portland cement and water in a weight ratio of 3:1.5:1:0.7. An uncured concrete slurry is then injected into a 1.27 cm × 1.27 cm × 2.54 cm mold, stirred to remove gas bubbles, and cured for 3 days in ambient conditions. A partially cured concrete bar is dipped in an aqueous dispersion of α-$Fe_2O_3$ (0.25 g mL$^{-1}$) for 3 s, then dried in air. Synthesis is performed at 150 °C for 14 h in a 125 mL Teflon-lined stainless-steel autoclave loaded with 1 mL of a 0.45 M EDOT in chlorobenzene solution and 0.1 mL of 12 M HCl.

**Fabrication of a quasi-solid-state supercapacitor**. The brick is attached to Pt as shown in Supplementary Fig. 7a (left top) prior to gel casting. The gel electrolyte (0.1 g mL$^{-1}$ poly(vinyl alcohol)/1 M $H_2SO_4$) is pipetted onto two 1 cm × 0.5 cm × 0.28 cm PEDOT-coated bricks (100 μL each brick on the 1 cm × 0.5 cm face). This electrolyte is allowed to impregnate for 12 h at ambient conditions (25 °C, 30–60% relative humidity) forming a semidry gel layer. An additional 25 μL of gel electrolyte is added serving as a binder between two bricks to assemble a sandwich-type supercapacitor. A device is dried in ambient conditions (25 °C, 30–60% relative humidity) for 1 h before sealing with epoxy (Supplementary Fig. 9a).

**Fabrication of a tandem supercapacitor**. The gel electrolyte is a mixture of 0.1 g of poly(vinyl alcohol) in 1 mL of 1 M $H_2SO_4$ and 200 μL are added to a PEDOT-coated brick face (1 cm × 1 cm). This electrolyte is allowed to stabilize for 12 h at ambient conditions (25 °C, 30–60% relative humidity); this process is repeated for all brick electrodes resulting in a thick gel layer (Supplementary Fig. 11a). An additional 50 μL of poly(vinyl alcohol)/$H_2SO_4$ solution is added serving as a binder between bricks. Devices are dried in an oven at 50 °C for 2 h and platinum foil, serving as a lead, is attached to bricks using polyimide tape.

## Data availability
The data that support the findings of this study are available from the corresponding author on request.

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

## Acknowledgements
We thank Dr. Bryce Sadtler from the Department of Chemistry and Dr. Anne M. Hofmeister from the Department of Earth and Planetary Sciences at Washington University for providing fruitful discussions. We are grateful to Dr. Huafang Li from the Institute of Materials Science & Engineering at Washington University for helping with electron microscopy. We also acknowledge helpful technical input from Dr. Luciano M. Santino and Mr. Haozhe Chen and are grateful to Dr. G.S. D'Arcy for editing this work.

## Author contributions
H.W. and J.M.D. designed the experiments. H.W. carried out the synthesis, device fabrication and characterization. Q.Z. performed inductively coupled plasma mass spectrometry characterization and analyses. H.W. and J.M.D. analyzed data and wrote the paper. H.W., Y.D., Y.L., H.Y., Q.Z., K.C. and J.M.D. contributed to the discussion and editing of the paper.

## Competing interests
The authors declare no competing interests.
