## [Peer Review File · Nature Communications]

Reviewers' Comments:

Reviewer #1:

Remarks to the Author:

The manuscript appears technical sound, and is an interesting read. The concept of using bricks to store electrical energy is certainly novel. However, I do have concerns:

1. Could enough energy be stored by bricks to be of any possible use? Perhaps the authors could present a calculation as to the energy stored per unit area of a brick wall? Or simply admit the energy stored will be very small, and most suitable for powering some sort of embedded microdevice.
2. It was hard to determine what was the significant advance in the flexible supercapacitor. While the work appeared technically sound, it seemed rather similar to a number of other reports.

Given the interest likely by the general public in "storing energy in bricks", I suppose this is an worthwhile paper to publish. In the long run however, I suspect this paper will not add considerably to the field.

Reviewer #2:

Remarks to the Author:

I am afraid I've got a mitigated feeling about this manuscript. On one hand, the novelty seems to be very high and the presented results are quite impressive and innovative as well. On the other hand, after careful reading of the paper and of the supplementary materials I cannot determine if the authors are intentionally hiding the main parameters in a forest of data or if they are simply not aware of good practices in the field of electrochemical capacitors (same for batteries). What is most puzzling me is the choice they made to report all their data as areal values (F/cm², Wh/cm², etc.) while this only applies for microdevices or wearable electronics (see good practice in the field from ("A Guideline for Reporting Performance Metrics with Electrochemical Capacitors: From Electrode Materials to Full Devices", A. Balducci et al., J. Electrochem. Soc. 2017 volume 164, issue 7, A1487-A1488). In their case, the right metrics to be used are F/cm³ and F/g and so on. Moreover, I will say that if such values were provided instead of unrealistic surface values, I would not ask me the question whether this paper deserves to be published in Nature Comm. Or not and my answer would have been yes. The authors must be aware that the readers must have the correct information about electrode geometry at the right place which means in the manuscript together with the related electrochemical tests. In this version of the paper it seems geometries, shapes and electrochemical data are randomly split between the manuscript and the SI so that the reader cannot make adequate calculations for single electrodes, single device and series devices. It is mandatory to give the right information at the right place and not drown the pertinent data in a bunch of minor information. But the way it is presented opens up many questions that should probably prevent publication. In-depth revisions of the paper are required before it can be considered again for publication in Nature Comm. My second main concern is about the changes occurring to the bricks: what happened to Al₂O₃ or SiO₂? There seems to be no evidence of how these minerals further behave inside the brick. The data are required.

Additional comments (related or not to the two main comments are:

- 1) Line 73 & 74: concentrations values seem to come out of nowhere. What is the driving forces and limitations behind that?
- 2) Same question for 14h line 82?
- 3) Line 91: no news from other minerals from the brick?
- 4) Line 103: problems begin. Which surface are we talking about. What is the electrode volume? What is the electrode mass?
- 5) Line 104: the power density seems to be very low. A cycling rate of 2 mV/s is quite unusual in ECs

field. This point needs to be addressed in the manuscript.

6) Line 107: what is "Faradic pseudocapacitance". I know "pseudocapacitance" and "Faradic electrodes" but both terms combined together looks weird. Reading of ("To Be or Not To Be Pseudocapacitive?", D. Bélanger et al., J. Electrochem. Soc. 2015 volume 162, issue 5, A5185-A5189 can be of help.

7) Line 120: 3 Ohm is quite a huge value which will hinder power capability.

8) Line 124: problems continue. Which surface are we talking about. What is the device volume? What is the device mass? Is the value calculated for a single device electrode or is it the capacitance of the full device?

9) Line 132: it works up to 1.2V at the expense of ohmic drop and probably long term cycling ability.

10) Line 139: what are the device geometry and dimensions? A scheme inside the main text is required.

11) Line 159: what such temperature range? Any reason for not going higher?

12) Line 163: the waterproof module is still a mystery to me due to lack of clear explanations about its design.

13) Line 181: the abusive numbers of minerals tested is probably a source of misunderstanding of the scientific message. Most of the SI data are definitely not useful for the readers but deserves to be gathered in another paper and submitted elsewhere. It is clear for the reviewer that a lot of work has been done on bricks and this is largely enough to be considered for publication in Nature Comm.

14) Line 182: 6,194 mF/cm² (cm² of what?) together with 205 F/g of PEDOT (and not per g of fabric + PEDOT) do not inform the reader about gravimetric and volumetric performance for the real life device made out of PEDOT and carbon cloth.

15) Line 285 and whole paragraph: give these values in the related figure captions together with gravimetric and volumetric capacitances.

Finally, I would like to add that I tried hard to made some calculations according to the manuscript, the SI and Fig. 1 and 4. Sometimes interesting values came out of these calculations and sometimes I ended up with capacitance poor capacitance value (and a headache. I strongly doubt that the authors are not aware that they are not providing values with the right metric. They already did so in one of their previous paper in the field (Self-woven nanofibrillar PEDOT mats for impact-resistant supercapacitors, Sustainable Energy & Fuels, 2019). In this paper they provide gravimetric values without providing the mass loading of each electrode neither for the device (mass loading which is suspected to be very low).

Once again, the science and technology behind the current paper seem nice but they are not highlighted as they should be.

Reviewer #3:

Remarks to the Author:

In this manuscript, the authors developed a chemical synthesis to control oxidative radical polymerization and deposition of a nanofibrillar coating of the conducting polymer poly(3,4-ethylenedioxythiophene) (PEDOT) on fired brick and evaluated its capacitance performance based on this synthesis. The results are interesting, but some key issues still need to be addressed. The manuscript is recommended to be published after major modification. Here are my comments:

1. One of the highlights of this manuscript is using the iron oxide in bricks to induce polymerization reaction. But bricks are building materials after all. What is the purity of iron oxide in bricks? What's the size of the α -Fe₂O₃? Is it evenly distributed? Can different bricks guarantee the stability of quality? Does the conditions mentioned above affect the polymerization of PEDOT and the performance of capacitor? This is very important and needs to be characterized and evaluated in detail.

2. In this manuscript, bricks, natural minerals and rocks, and α -Fe₂O₃-impregnated carbon cloth are used as substrates to produce flexible nanofibrillar PEDOT. Is there any difference between PEDOT nanofibrillar made on bricks, other minerals and carbon cloth carrying iron oxide? Like size, polymerization degree, conductivity and so on? And their influences on capacitor performance needs to be analyzed.

3. As stated in the article, water evaporation from the gel's hydrated ionic percolation network effects the performance of quasi-solid-state nanofibrillar PEDOT-coated brick supercapacitor. How to ensure that the water content of the gel electrolyte is consistent for each device assembling?

4. What is the difference of areal capacitance of bricks with different volume? Does it increase linearly or remain unchanged?

5. Iron oxide itself has pseudo-capacitance properties. Since α -Fe₂O₃ nanoparticles are widely dispersed in bricks or α -Fe₂O₃-impregnated carbon cloth, does the capacitance performance of iron oxide reflect in this device? Meanwhile, is there any other metal oxide in the bricks? Does it affect capacitance performance? This is very important and directly affects the interpretation of capacitor performance.

6. How to explain the Nyquist plot in low-frequency range of symmetric nanofibrillar PEDOT-coated brick supercapacitor? (Supplementary Fig. 5e)

Reviewer 1

“The manuscript appears technical sound, and is an interesting read. The concept of using bricks to store electrical energy is certainly novel.”

Question 1: *“Could enough energy be stored by bricks to be of any possible use? Perhaps the authors could present a calculation as to the energy stored per unit area of a brick wall? Or simply admit the energy stored will be very small, and most suitable for powering some sort of embedded microdevice.”*

Answer: We thank the reviewer for this idea and as suggested, we have performed a calculation to estimate the capacitance and energy stored per unit area of a brick wall that demonstrates the viability for powering embedded microdevices. A brick wall model could deliver a maximum capacitance of 11,523 F/m² and an energy density of 1.606 Wh/m². To illustrate the brick wall model, we have created a new figure as Supplementary Fig. 12 and shown all calculations in a newly added Supplementary Discussion section. The results from our calculations are also indicated in the manuscript body (Page 11, line 244 and Page 12, line 254).

Question 2: *“It was hard to determine what was the significant advance in the flexible supercapacitor. While the work appeared technically sound, it seemed rather similar to a number of other reports.”*

Answer: After careful evaluation of the manuscript and after taking into account the comments from another reviewer, we agree that the story of “storing electricity in bricks” ought to be communicated as a singly focused narrative to maximize its impact. The flexible supercapacitor section has been removed from the manuscript and we have streamlined the focus of the manuscript by also removing mineral data as well as by developing a more in-depth study of PEDOT-coated bricks for energy storage. Newly added sections include: synthesis of polymer using different types of bricks (Fig. 2a,b and Page 5, line 93), powder X-ray diffraction analyses (Fig. 2c,d and Page 5, line 96), current-voltage analyses (Fig. 3a and Page 5, line 106), water absorption tests (Fig. 3b and Page 5, line 110), Scotch tape tests (Fig. 3c and Page 5, line 113), adhesion analyses (Fig. 3d-f and Page 5, line 117) and ICP-MS analysis of electrolyte (Supplementary Fig. 6b,c and Page 7, Line 158).

We thank the reviewer for taking the time to review our manuscript in detail and we sincerely appreciate all comments. We have strived to answer all questions and our manuscript is much stronger for it, thank you!

Reviewer 2

“The novelty seems to be very high and the presented results are quite impressive and innovative as well.”

Main question 1-1: “What is most puzzling me is the choice they made to report all their data as areal values (F/cm^2 , Wh/cm^2 , etc.) while this only applies for microdevices or wearable electronics (see good practice in the field from (“A Guideline for Reporting Performance Metrics with Electrochemical Capacitors: From Electrode Materials to Full Devices”, A. Balducci et al., *J. Electrochem. Soc.* 2017 volume 164, issue 7, A1487-A1488). In their case, the right metrics to be used are F/cm^3 and F/g and so on.”

Answer: According to the paper provided above, F/cm^3 and F/g are “the most critical parameters” for “conventional electrochemical capacitors (EC)” because “for the final user, the weight and volume of such ECs are key parameters when choosing products”. Our stationary supercapacitor is not conventional by any stretch of the imagination and not designed for portable electronics or electric vehicles. The weight of our final device i.e., a brick wall, is less important than area and volume. When describing our brick wall device, we believe that areal energy metrics will readily communicate the quantitative relationship between laying bricks and storing energy given that area is the most common conventional unit for scaling wall construction.

Moreover, precedent exists for using areal metrics as a common parameter for evaluating thick electrodes that are neither micro sized nor used for wearable purposes (*Joule* 2019, 3, 459; *Adv. Mater.* 2018, 30, 1703027; *Electrochim. Acta* 2019, 302, 38; *Nat. Commun.* 2018, 9, 2578; *Nat. Energy* 2019, 4, 560; *Nature* 2018, 557,409).

To demonstrate the advantage of analyzing our data with areal metrics, we added a new figure correlating gravimetric, volumetric and areal metrics with the size of a brick electrode and device (Supplementary Fig. 5). Experimental data (Supplementary Fig. 9h) demonstrates that a device’s capacitance and energy are proportional to the contact area between PEDOT and electrolyte; a new figure illustrating a clear relationship between wall device and areal metrics has been added as Supplementary Fig. 12. Also, step-by-step calculations are included in a new Supplementary Discussion section.

For the purpose of completeness, we also calculated gravimetric and volumetric figures of merit for all of our electrodes and devices and have added Supplementary Table 1 (summary of dimension, volume, weight and mass loading for all electrodes and devices) and Supplementary Table 2 (summary of all areal, gravimetric and volumetric figures of merit). Moreover, we present step-by-step calculations in a newly added Supplementary Calculations section and address our reasoning for choosing areal metrics on Page 6, Line 126.

We present comprehensive calculations of figures of merits normalized under various metrics and facilitate evaluation and analysis of results. This new information is added on Page 6, Line 128.

Main question 1-2: *“The authors must be aware that the readers must have the correct information about electrode geometry at the right place which means in the manuscript together with the related electrochemical tests.”*

Answer: As suggested, we have modified our manuscript and it now presents all electrode geometries and devices alongside with related electrochemical tests and illustrations. We have added information about electrode geometry for three-electrode tests on Page 6, Line 133 and in the caption of Fig. 4a. We have added information about electrode geometry for an aqueous electrolyte supercapacitor on Page 8, Line 165 and caption of Supplementary Fig. 7a. We have added information about electrode geometry for a quasi-solid-state electrolyte supercapacitor on Page 9, Line 201 and caption of Supplementary Fig. 9a. We have summarized information for all electrodes and devices in a newly added Supplementary Table 1.

Main question 2: *“My second main concern is about the changes occurring to the bricks: what happened to Al_2O_3 or SiO_2 ? There seems to be no evidence of how these minerals further behave inside the brick. The data are required.”*

Answer: Great question! We apply powder X-ray diffraction for analyzing inorganic species in a brick electrode before and after electrochemical cycling. Results show that Al_2O_3 and Si_2O_3 are unaffected by both synthesis and electrochemical cycling. A brick electrode's X-ray diffraction pattern remains unchanged after repeated CV cycles collected at increasing scan rates of 2, 5, 10, 25, 50, 100 mV/s (10 cycles each under a -0.2 V to 0.8 V voltage window vs. Ag/AgCl and using 1 M H_2SO_4 as electrolyte). We have created a new section that discusses our findings (from Page 7, Line 152) and added a new figure panel to show data (Fig. 4f).

To confirm that Al_2O_3 and $\alpha-Fe_2O_3$ are not dissolved by the acidic electrolyte, we have analyzed the metal content in the electrolyte after cycling via inductively coupled plasma mass spectrometry which shows negligible concentrations of either Fe or Al in the electrolyte. We have added a new figure to summarize our findings in Supplementary Fig. 6 and address our results in Page 7, Line 158.

The lack of metal species in the electrolyte plausibly stems from our voltage window, -0.2 V to 0.8 V vs. Ag/AgCl (0.03 V to 1.03 V vs. SHE), as the standard reduction potentials (vs. SHE) are: $E_{SiO_2/Si} = -0.91$ V, $E_{Fe(III)/Fe(II)} = 0.77$ V and $E_{Al(III)/Al} = -1.662$ V. Therefore, $\alpha-Fe_2O_3$ is likely the only active species during our cycling.

Additional question 1: *“Line 73 & 74: concentrations values seem to come out of nowhere. What are the driving forces and limitations behind that?”*

Answer: The driving forces behind PEDOT's formation are dissolution of $\alpha\text{-Fe}_2\text{O}_3$ and oxidative radical polymerization. Our polymerization strategy minimizes acid-catalyzed polymerization by controlling acid concentration.

We have created a new section that explains why acid concentration matters and how concentration values are calculated (Page 3, Line 66). Our new description explains an acid's effect on oxidative radical polymerization and acid-catalyzed polymerization.

We have also added a newly modified scheme that addresses how acid concentration directs mechanistic (Fig. 1c).

Additional question 2: *“Same question for 14h line 82?”*

Answer: Kinetics control reaction mechanisms and our process is comprised of multiple reaction mechanism involving vaporization of reactants, $\alpha\text{-Fe}_2\text{O}_3$ dissolution, Fe^{3+} hydrolysis and EDOT polymerization. Extended syntheses at high temperatures lead to dopant loss as mentioned on Page 4, Line 85 and we have added a new section that explains the reason for controlling reaction time (Page 4, Line 79).

Additional question 3: *“Line 91: no news from other minerals from the brick?”*

Answer: We carried out powder X-ray diffraction on pulverized brick samples to elucidate minerals composition and found SiO_2 as the major phase while $\alpha\text{-Fe}_2\text{O}_3$ and Al_2O_3 are present as minor phases. Effects from synthesis and electrochemical cycling on brick and electrolyte were studied via powder X-ray diffraction and inductively coupled plasma mass spectrometry. X-ray diffraction shows that brick minerals are inert during both synthesis and cycling leading and inductively coupled plasma mass spectrometry shows negligible concentrations of Fe^{3+} and Al^{3+} in the electrolyte after cycling. We have added a new figure that shows the powder X-ray diffraction analysis of inorganic species in a brick as Fig. 2c and a new explanation of our results on Page 5, Line 96. We have also added a new Fig. 4f comparing powder X-ray diffraction patterns before synthesis and after cycling and its corresponding explanation (Page 7, Line 152). The results from inductively coupled plasma mass spectrometry are summarized in a newly added Supplementary Fig. 6 and their analysis is found on Page 7, Line 158.

Additional question 4: *“Line 103: problems begin. Which surface are we talking about. What is the electrode volume? What is the electrode mass?”*

Answer: To address this concern, we have added the following new information: 1) A section of step-by-step calculations for all values from electrochemistry enabling readers to perform calculations for themselves. 2) Two new tables summarizing geometries, dimensions, masses and mass loadings for all electrodes and devices as well performance based on gravimetric, volumetric and areal metrics.

We use the area of the electrode surface exposed to the electrolyte (0.5 cm^2) for calculations. The electrode volume is 0.14 cm^3 with a mass of 249 mg containing 6.97 mg PEDOT. We have summarized the electrode/device information in Supplementary Table 1 with calculations shown in Supplementary Calculations and summarized in Supplementary Table 2. In the manuscript, we have added electrode information on Page 6, Line 125.

Additional question 5: *“Line 104: the porosity density seems to be very low. A cycling rate of 2 mV/s is quite unusual in ECs field. This point needs to be addressed in the manuscript.”*

Answer: Our work contains scan rates ranging from 2 to 100 mV/s and we have thoroughly analyzed areal capacitances at different scan rates in three-electrode and two-electrode setups. Our goal was to provide the reader with an overall picture of the performance for all our electrodes and devices. We have summarized capacitances for three-electrode characterization at 2, 5, 10, 25, 50, 100 mV/s and presented data in a newly added Supplementary Fig. 6a; capacitances for two-electrode devices at different scan rates are summarized in our new Supplementary Fig. 7g (for aqueous electrolyte supercapacitor) and Supplementary Fig. 9e (for quasi-solid-state supercapacitor). Moreover, we have summarized device capacitances obtained from galvanostatic charge-discharge measurements at different current densities and plotted areal capacitances vs. current densities for aqueous electrolyte supercapacitor in a new Supplementary Fig. 7h, and for quasi-solid-state supercapacitor in a new Supplementary Fig. 9f. We have also added Ragone plots showing energy density vs. power density for aqueous electrolyte supercapacitor in a new Supplementary Fig. 7i, and for quasi-solid-state supercapacitor in Supplementary Fig. 9g that we believe will help the reader to rapidly compare all power densities against energy densities for all devices. We would like to point out to the reviewer that in the manuscript, we use low scan rates such as 2 mV/s to study electrochemical behaviors of the thick electrode and elucidate the maximum capacitance and energy density. We have added a sentence in the text that addresses this point on Page 7, Line 138.

Additional question 6: *“Line 107: what is “Faradic pseudocapacitance”. I know “pseudocapacitance” and “Faradic electrodes” but both terms combined together looks weird. Reading of (“To Be or Not To Be Pseudocapacitive?”, D. Bélanger et al., J. Electrochem. Soc. 2015 volume 162, issue 5, A5185-A5189 can be of help.”*

Answer: Thank you for pointing this out. The phrase “Faradaic pseudocapacitance” refers to pseudocapacitance contributed by Faradaic processes instead of ion intercalation. We adopted it from “Gogotsi Y, Penner RM. *Energy Storage in Nanomaterials - Capacitive, Pseudocapacitive, or Battery-like? ACS Nano* **12**, 2081-2083 (2018).” After a more thorough literature search, we discovered that this terminology is used with low frequency and we therefore have changed it to “Faradaic processes” on Page 7, line 143 to avoid any confusion.

Additional question 7: *“Line 120: 3 Ohm is quite a huge value which will hinder power capability.”*

Answer: We agree that a high internal resistance is harmful to power capability, thus we have created new plots to evaluate the rate capability and power performance of our electrode (Supplementary Fig. 6a), aqueous electrolyte supercapacitor (Supplementary Fig. 7g-i) and quasi-solid-state supercapacitor (Supplementary Fig. 9e-g).

We would like to point out that: 1) The gel electrolyte decreases the internal resistance of our supercapacitor to 2.5 Ohm (Supplementary Fig. 9c and Page 9, Line 208). 2) For electrochemical capacitors based on PEDOT electrodes, our 3 Ohm’s internal resistance is low compared to other reports (10 Ohm from “*Nat. Commun.* 2018, 9, 2578” and 5 Ohm from “*Energy Environ. Sci.* 2015, 8, 1339”).

Additional question 8: *“Line 124: problems continue. Which surface are we talking about. What is the device volume? What is the device mass? Is the value calculated for a single device electrode or is it the capacitance of the full device?”*

Answer: We have modified the paper by summarizing geometries, sizes, areas, masses and mass loadings of all our electrodes and devices in a newly added Supplementary Table 1. We have also included gravimetric, volumetric and areal metrics of all our electrodes and devices in another Supplementary Table 2.

We have also added new sections as Supplementary Equations and Supplementary Calculations to show all equations and step-by-step calculations for all metrics (gravimetric, volumetric and areal).

We utilized the electrode area directly in contact with the separator (0.5 cm²) as the device area for our calculations. The device volume is 0.28 cm³ and is characterized by 499 mg in weight (containing 13.94 mg of PEDOT). All values calculated in the device sections are for the full device and we indicated this on Page 2, Line 41; Page 8, Line 176; Page 9, Line 184; Page 10, Line 210; Page 10, Line 245.

Additional question 9: “Line 132: it works up to 1.2V at the expense of ohmic drop and probably long term cycling ability.”

Answer: We agree and our calculations from both cyclic voltammograms and galvanostatic charge-discharge profiles show that capacitances of aqueous electrolyte supercapacitor at 1.2 V are lower than 1 V due to the higher ohmic drop. However, the energy densities at 1.2 V are higher than 1 V because more charges are stored under a wider voltage window, as shown by a newly added Ragone plot (Supplementary Fig. 7i). We have summarized our data and added three new plots to compare the performance between 1 V and 1.2 V (Supplementary Fig. 7g-i). In the manuscript, we have also addressed the possible drawbacks of 1.2V voltage window including higher ohmic drop, lower coulombic efficiency, shorter cycle life and lower capacitance on Page 9, Line 190.

Additional question 10: “Line 139: what are the device geometry and dimensions? A scheme inside the main text is required.”

Answer: We have added a table that summarizes geometries and dimensions for all our electrodes and devices as Supplementary Table 1. Note that a quasi-solid-state device is a 1 cm × 0.5 cm × 0.63 cm cuboid containing two brick electrodes (1 cm × 0.5 cm × 0.28 cm) separated by a 0.7 mm thick gel electrolyte layer and sealed by an epoxy. We have added this scheme on Page 9, Line 200 and captions for Supplementary Fig. 9a.

Additional question 11: “Line 159: what such temperature range? Any reason for not going higher?”

Answer: We have three reasons for choosing our temperature range (-20 °C to 60 °C) to test the environmental stability of our stationary supercapacitor. 1) This range is typically utilized for testing most common rechargeable batteries (https://batteryuniversity.com/learn/article/charging_at_high_and_low_temperatures) and -20 °C is a common lower limit for supercapacitor temperature tests (e.g., *Nano Energy* 2014, 8, 231). 2) Between 20 °C and 60 °C, the vapor pressure of water increases at a rate of 0.05 MPa per 10 °C. Above 60 °C, this rate quadruples to ~ 0.2 MPa per 10 °C resulting in significant electrolyte water evaporation (*Biological and Bioenvironmental Heat and Mass Transfer, 2002, Chapter 7, Figure 7.16, page 136*). 3) This range covers most of outdoor temperatures which our wall device would experience. According to the World Meteorological Organization (WMO), 56.7 °C is the highest weather temperature ever recorded. We have added an explanation for this temperature range selection on Page 10, Line 225 and Page 11, Line 229.

Additional question 12: *“Line 163: the waterproof module is still a mystery to me due to lack of clear explanations about its design.”*

Answer: We have created a section that describes the waterproof module comprised of three quasi-solid-state supercapacitors connected in series and sealed in an epoxy puck (Supplementary Fig. 9a). We have added a new description of the design on Page 11, Line 232 and modified the illustration of the electrical circuit in Fig. 5e inset and Supplementary Fig. 10e. The entire module is coated epoxy leaving only leads for cathode and anode uncoated. After epoxy cures, the entire module is immersed in water (except for leads), electrochemical cycled and utilized for lighting a light-emitting diode.

Additional question 13: *“Line 181: the abusive numbers of minerals tested is probably a source of misunderstanding of the scientific message. Most of the SI data are definitely not useful for the readers but deserves to be gathered in another paper and submitted elsewhere. It is clear for the reviewer that a lot of work has bene done on bricks and this is largely enough to be considered for publication in Nature Comm.”*

Answer: We agree and therefore have streamlined the focus of the manuscript by removing the flexible supercapacitor data and mineral data. Our manuscript is more focused and newly added contents include the synthesis of polymer using different types of bricks (Fig. 2a,b and Page 5, line 93), powder X-ray diffraction analyses (Fig. 2c,d and Page 5, line 96), current-voltage analyses (Fig. 3a and Page 5, line 106), water absorption tests (Fig. 3b and Page 5, line 110), Scotch tape tests (Fig. 3c and Page 5, line 113), adhesion analyses (Fig. 3d-f and Page 5, line 117) and ICP-MS analyses of electrolyte (Supplementary Fig. 6b,c and Page 7, Line 158).

Additional question 14: *“Line 182: 6,194 mF/cm2 (cm2 of what?) together with 205 F/g of PEDOT (and not per g of fabric + PEDOT) do not inform the reader about gravimetric and volumetric performance for the real life device made out of PEDOT and carbon cloth.”*

Answer: Note that in order to streamline and highlight the most impactful areas of our work, we have removed the section regarding flexible supercapacitor research from our current manuscript. The area used for capacitance calculation is the electrode area i.e., $1.27 \text{ cm} \times 0.635 \text{ cm} = 0.806 \text{ cm}^2$ as mentioned in Methods section in the previous manuscript version.

Additional question 15: *“Line 285 and whole paragraph: give these values in the related figure captions together with gravimetric and volumetric capacitances.”*

Answer: Thank you for pointing this out! We have added the brick dimensions in the caption of Fig. 4a and Supplementary Fig. 7a and 9a. We have also added these

values in the text, along with gravimetric and volumetric capacitances on Page 6, Line 133; Page 8, Line 166; Page 9, Line 201; Page 11, Line 242.

Additional question 16: *“I strongly doubt that the authors are not aware that they are not providing values with the right metric. They already did so in one of their previous paper in the field (Self-woven nanofibrillar PEDOT mats for impact-resistant supercapacitors, Sustainable Energy & Fuels, 2019). In this paper they provide gravimetric values without providing the mass loading of each electrode neither for the device (mass loading which is suspected to be very low).”*

Answer: We appreciate the reviewer’s comment and in this paper we have added a new section and a new figure that explains why we use areal metrics to evaluate our unconventional supercapacitor (Page 6, Line 128 and Supplementary Fig. 5). We have also added geometries, dimensions, areas, masses and mass loadings for all our electrodes and devices in Supplementary Table 1, together with calculated gravimetric, volumetric and areal metrics in Supplementary Table 2 and step-by-step calculations shown in Supplementary Calculations. Moreover, we have modified our text and figure captions by reporting electrode/device geometries alongside with electrochemical data.

We would like to point out to the reviewer that in the first paragraph from Results and Discussion in the *“Self-woven nanofibrillar PEDOT mats for impact-resistant supercapacitors”* paper, we stated: “For both three-electrode and two-electrode characterization, the active electrode area is 40.3 mm^2 ($6.35 \times 6.35 \text{ mm}$) with an electrode mass of 0.25 mg and a mass density of 0.27 g cm^{-3} ”. In the second paragraph of Results and Discussion (“Design and merits of horizontally directed 1D PEDOT nanostructures”) we also provided the thickness of the electrode (23 micrometers thick) for volumetric calculation.

We thank the reviewer for taking the time to review our manuscript in detail and we sincerely appreciate all comments. We have strived to answer all questions and our manuscript is much stronger for it, thank you!

Reviewer 3

“The results are interesting, but some key issues still need to be addressed. The manuscript is recommended to be published after major modification.”

Question 1-1: *“One of the highlights of this manuscript is using the iron oxide in bricks to induce polymerization reaction. But bricks are building materials after all. What is the purity of iron oxide in bricks?”*

Answer: We utilized X-ray powder diffraction to answer this question and pulverized bricks to obtain a homogenized fine powder for analysis. Data shows SiO_2 as

the major phase and α -Fe₂O₃ and Al₂O₃ as minor phases. We have created a new figure that illustrates these results (Fig. 2c) and modified the text accordingly (Page 5, Line 96).

Question 1-2: *“What's the size of the α -Fe₂O₃?”*

Answer: We performed scanning electron micrographs and optical micrographs on bricks to answer this question. Unfortunately, brick particles are highly fused and characterization of a single α -Fe₂O₃ particle proved unsuccessful. However, scanning electron micrographs provided a rough estimation of particle size (approximately 10 microns). We have added a new Fig. 2a that summarizes microscopy results.

Question 1-3: *“Is it evenly distributed?”*

Answer: We have added optical micrographs of bricks that show macroscopic distribution of α -Fe₂O₃ where the orange-red color (due to α -Fe₂O₃) was evenly distributed. We carried out analysis on three different types of bricks and we believe that α -Fe₂O₃ homogeneously distributed throughout the brick microstructure. We have added new micrographs in Fig. 2a.

Question 1-4: *“Can different bricks guarantee the stability of quality? Does the conditions mentioned above affect the polymerization of PEDOT and the performance of capacitor? This is very important and needs to be characterized and evaluated in detail.”*

Answer: This is an excellent question! To test the synthesis on different bricks, we have purchased three different types of bricks (type 1, type 2 and type 3 brick) from: The Home Depot Inc., LOWES Inc. and Menard's Inc (details are added in Method section). We have performed powder X-ray diffraction analysis which shows similar inorganic content, as well as scanning electron microscopy and water absorption tests that show differences in pore size and pore volume. We have also added digital photos that show macroscopic differences in gravel (SiO₂) size. Scanning electron micrographs demonstrate that under the similar synthetic conditions, that all types of bricks produce homogeneous PEDOT nanofibers with similar aspect-ratios. We have measured the current-voltage profiles of the three PEDOT coatings and results show similar Ohmic behavior and electrical resistance. Interestingly, a Scotch-tape test demonstrates differences in PEDOT coating adhesion to a brick. This difference stems from differences in brick porosities that affect vapor diffusion during synthesis and control PEDOT adhesion on a brick surface.

We have found that type 1 brick, characterized by the highest porosity, shows greatest adhesion via Scotch tape test because after synthesis, an embedded PEDOT network is present in the brick matrix. The other two types of bricks (type 2 and type 3 bricks are characterized by low porosity that stifles vapor diffusion during synthesis), show partial PEDOT delamination during Scotch tape tests because they lack an integrated embedded polymer network

throughout the brick pores. As a consequence of low porosity, segregated PEDOT coatings form that lack anchoring and are prone to delamination.

We have summarized all data in new sections of the manuscript (Page 5, Line 93 to Page 6, Line 124) and created two new figures (Fig. 2 and Fig. 3).

Question 2: *“In this manuscript, bricks, natural minerals and rocks, and α -Fe₂O₃-impregnated carbon cloth are used as substrates to produce flexible nanofibrillar PEDOT. Is there any difference between PEDOT nanofibrillar made on bricks, other minerals and carbon cloth carrying iron oxide? Like size, polymerization degree, conductivity and so on? And their influences on capacitor performance needs to be analyzed.”*

Answer: After careful evaluation of our work, we have streamlined the focus of our findings by removing the flexible supercapacitor data and mineral data from this manuscript. Our new manuscript addresses our novelty by producing a more in-depth study of PEDOT-coated bricks for energy storage. Newly added contents include the synthesis of polymer using different types of bricks (Fig. 2a,b and Page 5, line 93), powder X-ray diffraction analyses (Fig. 2c,d and Page 5, line 96), current-voltage analyses (Fig. 3a and Page 5, line 106), water absorption tests (Fig. 3b and Page 5, line 110), Scotch tape tests (Fig. 3c and Page 5, line 113), adhesion analyses (Fig. 3d-f and Page 5, line 117) and ICP-MS analyses of electrolyte (Supplementary Fig. 6b,c and Page 7, Line 158).

Question 3: *“As stated in the article, water evaporation from the gel’s hydrated ionic percolation network effects the performance of quasi-solid-state nanofibrillar PEDOT-coated brick supercapacitor. How to ensure that the water content of the gel electrolyte is consistent for each device assembling?”*

Answer: We have modified the Methods section (Page 15, Line 341) and present a strategy for obtaining reproducible gel formulations by controlling water content. This entails monitoring of ambient relative humidity and temperature as well as careful control of annealing process. We also monitor the volume and concentration of diluted gel electrolyte applied on an electrode as well as the temperature and duration of drying during device assembly. Moreover, our epoxy seal prevents water evaporation after device assembly.

Question 4: *“What is the difference of areal capacitance of bricks with different volume? Does it increase linearly or remain unchanged?”*

Answer: The areal capacitance remains unchanged with different brick volumes because it is determined by the thickness and structure of the PEDOT coating. However, in an aqueous electrolyte-based supercapacitor, two sides of a PEDOT-coated brick electrode contribute to charge storage. Therefore in an aqueous electrolyte-based supercapacitor, decreasing the electrode volume decreases

areal capacitance. We demonstrate this effect by cutting off one the PEDOT-coated faces thereby decreasing the areal capacitance by half (Fig. 5a and Supplementary Fig. 9h).

We have newly added a figure (Supplementary Fig. 5) for the readers to understand clearly how the areal capacitance will change based on different brick volumes.

Question 5-1: *“Iron oxide itself has pseudo-capacitance properties. Since α -Fe₂O₃ nanoparticles are widely disputed in bricks or α -Fe₂O₃-impregnated carbon cloth, does the capacitance performance of iron oxide reflect in this device?”*

Answer: Thank you for pointing this out. We have previously shown that iron oxide contributes to the capacitance at 2 mV/s scan rate in three-electrode cyclic voltammetry leading to a pair of redox peaks. However, these peaks disappear when the scan rate increases to 25 mV/s as well as in two-electrode measurements because of the slow charging kinetics of α -Fe₂O₃. To make it clear to the readers, we have clarified the minimal contribution of α -Fe₂O₃ to device capacitances (Page 10, Line 213). In the case for the carbon cloth electrode, we indicated previously in the Methods section that α -Fe₂O₃ is thoroughly removed by acid wash after synthesis (the removal of α -Fe₂O₃ were also confirmed by thermogravimetric analysis), therefore the α -Fe₂O₃ does not contribute to the capacitance in the flexible supercapacitor.

Question 5-2: *“Meanwhile, is there any other metal oxide in the bricks?”*

Answer: We have added powder X-ray diffraction of pulverized brick to elucidate all the inorganic components. This new data shows SiO₂ as the major phase in the brick while α -Fe₂O₃ and Al₂O₃ as the minor phases. We have added a new figure and a new section to address these powder X-ray diffraction patterns in Fig. 2c and Page 5, Line 96.

Question 5-3: *“Does it affect capacitance performance? This is very important and directly affects the interpretation of capacitor performance.”*

Answer: We answer this question from both theoretical and experimental perspectives. Theoretically, we noticed that the standard reduction potentials (vs. SHE) of $E_{\text{SiO}_2/\text{Si}} = -0.91$ V, $E_{\text{Fe (III)/Fe (II)}} = 0.77$ V and $E_{\text{Al (III)/Al}} = -1.662$ V therefore α -Fe₂O₃ is more than likely the only active species during our cycling between -0.2 V to 0.8 V vs. Ag/AgCl (0.03 V to 1.03 V vs. SHE).

Experimentally, we only observed redox peaks from iron oxide via three-electrode cyclic voltammogram as shown previously (currently as Figure 4b). To investigate the effect of electrochemical cycling on SiO₂ and Al₂O₃, we used powder X-ray diffraction to analyze inorganic species in a brick before and after cycling. Results show that the Al₂O₃ and Si₂O₃ are unaffected by synthesis

and electrochemical cycling because a brick electrode exhibits unchanged X-ray diffraction patterns after being cycled from -0.2 V to 0.8 V (vs. Ag/AgCl) in 1 M H₂SO₄ at 2, 5, 10, 25, 50, 100 mV/s (10 cycles each). We have created a new section to discuss inorganic species after electrochemical cycling (Page 7, Line 152) and added a new figure panel with data (Fig. 4f). To confirm that Al₂O₃ and α -Fe₂O₃ are not dissolved by the acidic electrolyte, we have analyzed the metal contents in the electrolyte after cycling. Analysis via inductively coupled plasma mass spectrometry shows negligible Fe and Al concentrations. We have added a new figure to summarize these findings (Supplementary Fig. 6) and addressed the results on Page 7, Line 158.

Question 6: “How to explain the Nyquist plot in low-frequency range of symmetric nanofibrillar PEDOT-coated brick supercapacitor? (Supplementary Fig. 6e)”

Answer: In the low-frequency region of Nyquist plot, the curve forms an arc because of Nernst diffusion impedance of H⁺ and SO₄²⁻ in the electrolyte, indicating a separation gap between anode and cathode that increases ion diffusion distance. Similar Nyquist plots are found in PEDOT:PSS-based textile electrodes (*Materials* 2018, 11, 48), rGO/CuO/PEDOT supercapacitors (*Plastics, Rubber and Composites*, 2019, 48, 168), Bi₂WO₆-Graphene pseudocapacitors (*Sci. Rep.* 2015, 5, 8624), ITO electrodes (*Measurement* 2013, 46, 2411) and dye-sensitized solar cells (*J. Mater. Chem. A* 2013, 1, 3932). We have added a discussion on Page 8, Line 172.

We thank the reviewer for taking the time to review our manuscript in detail and we sincerely appreciate all comments. We have strived to answer all questions and our manuscript is much stronger for it, thank you!

Authors' Contact Information:

Prof. Julio M. D'Arcy
 Department of Chemistry
 Institute of Materials Science & Engineering
 Washington University in St. Louis
 One Brookings Drive, Campus Box 1134
 Saint Louis, Missouri 63130-4899
 Email: jdarcy@wustl.edu
 Tel: 314-935-7586; Fax: 314-935-6530

Mr. Hongmin Wang
 Department of Chemistry
 Washington University in St. Louis
 One Brookings Drive, Campus Box 1134
 Saint Louis, Missouri 63130-4899
 Email: hongminwang@wustl.edu
 Tel: 314-935-2944; Fax: 314-935-6530

Mr. Yifan Diao
 Institute of Materials Science & Engineering
 Washington University in St. Louis
 One Brookings Drive, Campus Box 1134
 Saint Louis, Missouri 63130-4899
 Email: yifandiao@wustl.edu
 Tel: 314-935-2944; Fax: 314-935-6530

Mr. Yang Lu
 Institute of Materials Science & Engineering
 Washington University in St. Louis
 One Brookings Drive, Campus Box 1134
 Saint Louis, Missouri 63130-4899
 Email: ylu29@wustl.edu
 Tel: 314-935-2944; Fax: 314-935-6530

Mr. Haoru Yang
 Department of Chemistry
 Washington University in St. Louis
 One Brookings Drive, Campus Box 1134
 Saint Louis, Missouri 63130-4899
 Email: yanghaoru@wustl.edu
 Tel: 314-935-2944; Fax: 314-935-6530

Mrs. Qingjun Zhou
 Institute of Materials Science & Engineering
 Washington University in St. Louis
 One Brookings Drive, Campus Box 1134
 Saint Louis, Missouri 63130-4899
 Email: qingjunzhou@wustl.edu
 Tel: 314-312-8583

Mr. Kenneth Chrulski
 Department of Chemistry
 Washington University in St. Louis
 One Brookings Drive, Campus Box 1134
 Saint Louis, Missouri 63130-4899
 Email: kenneth.chrulski@wustl.edu
 Tel: 314-935-2944; Fax: 314-935-6530

Sincerely,

Julio M. D'Arcy, Ph.D.
 Assistant Professor of Chemistry
 Washington University in St. Louis
 E-mail: jdarcy@wustl.edu
 Tel.: (314) 935-7586;
 Fax: (314) 935-4481

Reviewers' Comments:

Reviewer #1:

Remarks to the Author:

I'm satisfied with the revised manuscript, and find it much more suitable for Nature Communications (keeping the story simple by removing the flexible capacitor discussion certainly helped!). I suggest Fig. 5d be revised to plot the CE from 80-100% or 90-100%. The way it is plotted now makes it impossible to actually determine the CE.

Reviewer #2:

Remarks to the Author:

The reviewer acknowledges the impressive work the authors made for making their paper clearer and focused on the innovation using bricks. It is really a giant improvement to provide the geometric details of all the electrodes and devices tested in the manuscript. Most of my previous questions have been pertinently addressed and the corresponding changes in the manuscript are quite satisfying. I have some remaining comments that may help the authors to further improve their manuscript.

Main remarks:

1) I am a little bit puzzled by the answer provided by the authors to one of my question. Line 144: "The Fe³⁺/Fe²⁺ redox pair at 0.37 V and 0.49 V (versus Ag/AgCl) arise due to iron species in brick and disappears as scan rate increases to 25 mV/s because double-layer capacitance outcompetes Faradaic processes at faster rates". Am I just wondering if the authors mean PEDOT stores charges by a capacitive process. If so, this is not correct: Electronically Conducting Polymers such as PEDOT store charges by doping/dedoping (faradic), rather than adsorption/desorption (non-faradic). So at least the sentence must be corrected but the authors must also check all their assumption in the manuscript and in the SI about this fact. Figure 4c simply evidences that Faradaic processes in PEDOT are faster than those occurring from solid Fe₂O₃.

2) The authors state that: "Our stationary supercapacitor is not conventional by any stretch of the imagination and not designed for portable electronics or electric vehicles. The weight of our final device i.e., a brick wall, is less important than area and volume." This is clear now but wasn't in the original version, with too many data and too many devices. I am still convinced that volumetric energy and power densities must be implemented in Supplementary Table 2, and must appear somewhere when the use of brick walls is mentioned together with surface values. It seems that the authors are also convinced that volume matters as well when talking of a building, so why being reluctant to provide such values to the attention of the readers. SI is one option, but few word about this fact in the manuscript will also give this information to "fast readers" and probably provide more citations to the future paper.

Minor remarks:

1) The reported values, especially for capacitance, etc..., contain too many significant figures. See for example: "maximum device capacitance of 11,523 F/m² and an energy density of 1.606 Wh/m²". It is quite difficult to provide such accuracy and if the authors think their values are correct they must provide the error bars; I strongly doubt that they will give such accuracy and shortening the figures will give some additional credibility to the study.

2) The authors answered to my previous question about pseudocapacitance: "Thank you for pointing this out. The phrase "Faradaic pseudocapacitance" refers to pseudocapacitance contributed by Faradaic processes instead of ion intercalation. We adopted it from "Gogotsi Y, Penner RM. Energy Storage in Nanomaterials - Capacitive, Pseudocapacitive, or Battery-like? ACS Nano 12, 2081-2083 (2018)." After a more thorough literature search, we discovered that this terminology is used with low frequency and we therefore have changed it to "Faradaic processes" on Page 7, line 143 to avoid any

confusion." They must be aware that ion intercalation is a Faradaic process also. The term pseudocapacitance designates a Faradaic process that provides a capacitive-like signature to the electrode which is quite uncommon in electrochemical energy storage devices.

Reviewer #3:

Remarks to the Author:

The authors have revised the manuscript carefully and the questions have been well addressed. Therefore, the manuscript is recommended to be published.

Department of Chemistry

June 11, 2020

Research Manuscript Submission: “ α -Fe₂O₃-derived Stationary PEDOT Supercapacitors for Energy Storage in Bricks”

Dear Editor:

Here are Our point-by-point response to issues raised by referees.

Referee 1

Question 1: I suggest Fig. 5d be revised to plot the CE from 80-100% or 90-100%. The way it is plotted now makes it impossible to actually determine the CE.

Answer: We thank referee 1 for this suggestion. We have revised the CE range (Fig. 5d) from 80-100%.

Referee 2

Main remarks

Question 1: I am a little bit puzzled by the answer provided by the authors to one of my question. Line 144: “The Fe³⁺/Fe²⁺ redox pair at 0.37 V and 0.49 V (versus Ag/AgCl) arise due to iron species in brick and disappears as scan rate increases to 25 mV/s because double-layer capacitance outcompetes Faradaic processes at faster rates”. Am I just wondering if the authors mean PEDOT stores charges by a capacitive proves. If so, this is not correct: Electronically Conducting Polymers such as PEDOT store charges by doping/dedoping (faradic), rather than adsorption/desorption (non-faradic). So at least the sentence must be corrected but the authors must also check all their assumption in the manuscript and in the SI about this fact. Figure 4c simply evidences that Faradaic processes in PEDOT are faster than those occurring from solid Fe₂O₃.

Answer: We thank referee 2 for helping us clarify this sentence (page 7, line 145) and have modified it, as suggested, from “... because double-layer capacitance outcompetes Faradaic processes at faster rates...” to “...Faradaic processes in PEDOT are faster than those occurring from solid Fe₂O₃...”. We also checked the rest of the manuscript and SI to ensure compliance with the fact that conducting polymers store charges by a faradic mechanism.

Question 2: I am still convinced that volumetric energy and power densities must be implemented in Supplementary Table 2, and must appear somewhere when the use of brick walls is mentioned together with surface values. It seems that the authors are also convinced that volume matters as well when talking of a building, so why being reluctant to provide such values to the attention of the readers. SI is one option, but few word about this fact in the manuscript will also give this

information to “fast readers” and probably provide more citations to the future paper.

Answer: We appreciate the referee’s kind suggestion and agree that it makes our work more thorough and complete. As suggested, we have modified Supplementary Table 2 to include volumetric energy and power densities. Moreover, we have also added these values along with areal metrics in the body of the manuscript (page 9, line 189 and page 10, line 216).

Minor remarks

Question 1: The reported values, especially for capacitance, etc..., contain too many significant figures. See for example: “maximum device capacitance of 11,523 F/m² and an energy density of 1.606 Wh/m²”. It is quite difficult to provide such accuracy and if the authors think their values are correct they must provide the error bars; I strongly doubt that they will give such accuracy and shortening the figures will give some additional credibility to the study.

Answer: We thank the referee for this suggestion and have reduced the number of significant figures and modified manuscript on page 2, line 42; page 7, line 140; page 8, line 179; page 9, line 187; page 10, line 216 and Supplementary Fig. 12 caption; Supplementary Table 2; Supplementary Methods; Supplementary Discussion.

Question 2: The authors answered to my previous question about pseudocapacitance: “Thank you for pointing this out. The phrase “Faradaic pseudocapacitance” refers to pseudocapacitance contributed by Faradaic processes instead of ion intercalation. We adopted it from “Gogotsi Y, Penner RM. Energy Storage in Nanomaterials - Capacitive, Pseudocapacitive, or Battery-like? ACS Nano 12, 2081-2083 (2018).” After a more thorough literature search, we discovered that this terminology is used with low frequency and we therefore have changed it to “Faradaic processes” on Page 7, line 143 to avoid any confusion.” They must be aware that ion intercalation is a Faradaic process also. The term pseudocapacitance designates a Faradaic process that provides a capacitive-like signature to the electrode which is quite uncommon in electrochemical energy storage devices.

Answer: We thank the referee for helping us focus on correct terminology and fundamentals of pseudocapacitance and capacitance. Our paper has benefitted greatly, thank you!

Referee 3

We thank referee 3 for acknowledging our revision without proposing any more question.

Sincerely,

Wustl University, One Brookings Drive, Saint Louis, Missouri 63130-4899
(314) 935-6530 Fax: (314) 935-4481 <http://www.chemistry.wustl.edu>

Julio M. D'Arcy, Ph.D.
Assistant Professor of Chemistry
Washington University in St. Louis
E-mail: jdarcy@wustl.edu
Tel.: (314) 935-7586; Fax: (314) 935-4481